# Depletion of Mitochondrial Components from Extracellular Vesicles Secreted from Astrocytes in a Mouse Model of Fragile X Syndrome

**DOI:** 10.3390/ijms22010410

**Published:** 2021-01-02

**Authors:** Byung Geun Ha, Jung-Yoon Heo, Yu-Jin Jang, Tae-Shin Park, Ju-Yeon Choi, Woo Young Jang, Sung-Jin Jeong

**Affiliations:** 1Research Group of Developmental Disorders and Rare Diseases, Korea Brain Research Institute (KBRI), Daegu 41062, Korea; BGHa@kbri.re.kr (B.G.H.); ungikimi@kbri.re.kr (J.-Y.H.); yudy@kbri.re.kr (Y.-J.J.); tspark88@kbri.re.kr (T.-S.P.); chl8725@kbri.re.kr (J.-Y.C.); aonaaomo@naver.com (W.Y.J.); 2Department of Brain and Cognitive Sciences, Daegu Gyeongbuk Institute of Science and Technology (DGIST), Daegu 42988, Korea

**Keywords:** extracellular vesicles, mitochondrial dysfunction, astrocytes, Fragile X syndrome, *Fmr1 knockout* mouse

## Abstract

Mitochondrial dysfunction contributes to neurodegenerative diseases and developmental disorders such as Fragile X syndrome (FXS). The cross-talk between mitochondria and extracellular vesicles (EVs) suggests that EVs may transfer mitochondrial components as intermediators for intracellular communication under physiological and pathological conditions. In the present study, the ability of EVs to transfer mitochondrial components and their role in mitochondrial dysfunction in astrocytes were examined in the brains of *Fmr1 knockout (KO)* mice, a model of FXS. The amounts of mitochondrial transcription factor NRF-1, ATP synthases ATP5A and ATPB, and the mitochondrial membrane protein VDAC1 in EVs were reduced in cerebral cortex samples and astrocytes from *Fmr1 KO* mice. These reductions correspond to decreased mitochondrial biogenesis and transcriptional activities in *Fmr1 KO* brain, along with decreased mitochondrial membrane potential (MMP) with abnormal localization of vimentin intermediate filament (VIF) in *Fmr1 KO* astrocytes. Our results suggest that mitochondrial dysfunction in astrocytes is associated with the pathogenesis of FXS and can be monitored by depletion of components in EVs. These findings may improve the ability to diagnose developmental diseases associated with mitochondrial dysfunction, such as FXS and autism spectrum disorders (ASD).

## 1. Introduction

Fragile X syndrome (FXS), an inherited developmental disorder characterized by mental retardation and symptoms of autism spectrum disorders (ASD), is caused by transcriptional silencing of the *FMR1* gene, which encodes fragile X mental retardation protein (FMRP) [1]. FMRP is an RNA-binding protein that is expressed primarily in neurons and astrocytes of the brain and associated with approximately 4% of transcripts, including those for mitochondrial proteins [2]. Alterations of mitochondrial proteins result in mitochondrial dysfunction, which is associated with various neurodegenerative diseases and developmental disorders [3]. The brains of *Fmr1 knockout (KO)* mice, a model of FXS, exhibit increases in glucose metabolism [4,5], oxidative stress [6], and reactive oxygen species production as well as abnormal nitric oxide metabolism [7] and systemic energy metabolism [8]. Developing neurons from *Fmr1 KO* mice show impaired dendritic maturation, altered expression of mitochondrial genes, fragmented mitochondria, impaired mitochondrial function, and increased oxidative stress [9]. However, it is not known if astrocytes from *Fmr1 KO* mice similarly show mitochondrial dysfunction.

Mitochondria are transferred between cells under disease conditions such as stroke [10], cancer [11], and lung injury [12]. However, details on the mechanism of mitochondrial transfer remain elusive. Mitochondrion-derived vesicles or direct interorganelle contacts could represent a mechanism to rapidly relieve mitochondrial stress to maintain metabolism, particularly when other degradation pathways are compromised [13]. Mitochondrial components, including proteins and mitochondrial DNA, have been detected in extracellular vesicles (EVs) derived from mesenchymal stem cells [14], and cytochrome *c* oxidase (COX) subunit I (encoded by mitochondrial DNA) and COX6c are enriched in EVs derived from tissues and plasma of melanoma patients [15]. These studies suggest that mitochondrial components are secreted from the cells via EVs.

EVs are secreted by all cell types upon the fusion of a multivesicular body with the plasma membrane [16]. EVs harbor various tissue-specific and disease-related molecules, including cellular and mitochondrial DNA, RNAs, miRNA, lipids, and proteins. EVs are remarkably stable in body fluids, proving their utility for monitoring disease biomarkers. EVs are known to transfer pathogens such as prion protein [17] (responsible for Creutzfeldt–Jakob disease), α-synuclein (involved in the pathogenesis of Parkinson’s disease), and amyloid β [18] and phosphorylated tau [19] (deposited in the brains of Alzheimer’s disease patients). However, EVs also aid in the elimination of toxins and pathogens from cells and transfer beneficial molecules. Recent studies have shown that factors with neurotrophic and neuroprotective properties are released from astrocytes via EVs to promote neurite outgrowth and neuronal survival under conditions of neurotransmitter toxicitys [20,21].

In the present study, we explored the role of mitochondrial function in astrocytes from *Fmr1 KO* mice and whether EVs propagate mitochondrial proteins for intercellular communication. We found that the levels of mitochondrial components are reduced in mitochondrial fractions from cortical tissues and astrocytes of *Fmr1 KO* mouse brains. This depletion reflects a decrease in their expression in mitochondrial biogenesis. Intriguingly, we were able to monitor the reductions in mitochondrial components in EVs from these samples. These results suggest that astrocytic mitochondrial dysfunction is associated with the pathogenesis of FXS and that this can be monitored in EVs in this disease as well as other neurodegenerative disorders and ASD.

## 2. Results

### 2.1. Decreased Expression of Mitochondrial Components in Mitochondrial Fractions but Not Cortical Lysates from Fmr1 KO Mice

To explore the levels of mitochondrial components, we performed western blot analyses on cortical lysates and mitochondrial fractions (See also Appendix A) from mouse brain tissues. ATP synthase alpha-subunit (ATP5A) and beta-subunit (ATPB), COX subunit I encoded by mitochondrial DNA (MT-CO1), and voltage-dependent anion-selective channel 1 (VDAC1) were detected equally in cortical lysates from wild-type (WT) and *Fmr1 KO* mice, which patterns are similar to their transcript’s levels with no significance (See also Appendix A), but ATP5A and ATPB levels were significantly lower in mitochondrial fractions from *Fmr1 KO* mice (Figure 1).

### 2.2. Reduction in Mitochondrial Biogenesis and Transcriptional Activity in Cortices of Fmr1 KO Mice

To determine whether the observed reduction of mitochondrial components is associated with mitochondrial biogenesis and transcription, we investigated mtDNA content and expression of genes related to mitochondrial transcription such as MT-CO1 (*mt-Co1*) and 16S for mitochondrial biogenesis or nuclear factor, erythroid derived 2, like 1 (NRF-1; *Nfe2l1*), nuclear factor, erythroid derived 2, like 2 (NRF-2; *Nfe2l2*), and mitochondrial transcription factor A (mtTFA; *Tfam*). Quantitative real-time polymerase chain reaction (qRT-PCR) results revealed that the mitochondrial DNA levels of mt-Co1 and 16S were significantly decreased in cortical tissues of *Fmr1 KO* mice (Figure 2). 

Additionally, levels of *Nfe2l1*, *Nfe2l2,* and *Tfam* were significantly reduced in cortical tissues from *Fmr1 KO* mice compared with that in WT mice (Figure 3A). The reduced levels of transcripts resulted in decreased expression at the protein level in mitochondrial fractions, as revealed by western blot analysis with antibodies against anti-mtTFA (Figure 3B) and NRF-1 (See also Appendix A). These results suggest that the diminished expression of mitochondrial components in cortices of *Fmr1 KO* mice is associated with a deterioration of mitochondrial biogenesis and transcriptional activity.

### 2.3. Reduced Levels of Mitochondrial Proteins in Mitochondrial Fractions from Astrocytes of Fmr1 KO Mice

Mitochondrial dysfunction affects the migration, development, and survival of neurons in *Fmr1 KO* mice [9]. By contrast, loss of mitochondrial DNA or mitochondrial activity enhances the function of astrocytes [22], but it is not clear whether this is altered in FXS. To address this, we performed western blot analysis using mitochondrial fractions and lysates prepared from astrocytes of *Fmr1 KO* mice. Expression of MT-CO1 and ATPB was significantly higher in total lysates from *Fmr1 KO* astrocytes than in those from WT astrocytes, whereas the expression of ATP5A and VDAC1 was nearly the same in those of *Fmr1 KO* and WT astrocytes. However, in the mitochondrial fractions from astrocytes, their expression was drastically reduced (Figure 4). These results suggest that a loss of mitochondrial proteins from astrocytes in the brains of *Fmr1 KO* mice may be the result of a dysfunction in mitochondrial protein trafficking at the intracellular level.

### 2.4. Decreased Mitochondrial Membrane Potential (MMP) in Astrocytes from Fmr1 KO Mice

To determine if the reduction in mitochondrial proteins results in altered mitochondrial function, we assessed the mitochondrial membrane potential (MMP) in astrocytes. The changes in MMP were visualized via the fluorescence of the cationic dye tetraethyl benzimidazolyl carbocyanine iodide (JC-1). JC-1 is predominantly a monomer that yields green fluorescence at low concentrations (due to low mitochondrial membrane potential) but aggregates and fluoresces orange under high MMP conditions. The JC-1 orange/green fluorescence ratio was significantly lower in *Fmr1 KO* astrocytes than in WT astrocytes. Moreover, the levels of JC-1 aggregates (i.e., orange fluorescence) and JC-1 monomers (green fluorescence) were lower in astrocytes from *Fmr1 KO* mice than in those from WT mice (Figure 5A). These data suggest that the observed depletion of mitochondrial proteins contributes to the reduced levels of MMPs in astrocytes from *Fmr1 KO* mice. To investigate the distribution of vimentin intermediate filament (VIF), which maintains the MMP and stabilizes the intracellular location and activity of mitochondrial proteins, we performed double immunofluorescence staining with glial fibrillary acidic protein (GFAP) and VIF in astrocytes. In comparison to VIF in WT astrocytes, *Fmr1 KO* astrocytes showed condensed VIF in astrocyte endfeet (Figure 5B). These results suggest that the reduced level of MMP in *Fmr1 KO* astrocytes may be caused by the abnormal distribution of VIF. 

### 2.5. Mitochondrial Components are Depleted in EVs Secreted by Cortices and Astrocytes of Fmr1 KO Mice

As mitochondrial proteins have been detected in EVs purified from plasma [15], we examined whether mitochondrial dysfunction could also be monitored by observing depletion of mitochondrial components in EVs secreted from astrocytes. EVs were purified by buoyant density gradient-ultracentrifugation [23], and their shape and size distributions were confirmed by nanoparticle tracking analysis (NTA) and transmission electron microscopy (TEM). The expression of EVs makers such as CD9 and CD81 was also confirmed (see also Appendix A). We first examined EVs purified from cortices of 10-week-old WT or *Fmr1 KO* mice. Western blot analysis revealed that the levels of NRF-1, ATP5A, ATPB, and VDAC1 were dramatically lower in EVs from the cortices of *Fmr1 KO* mice than in those from WT controls (Figure 6A). We next examined EVs purified from the medium of astrocytes cultured for 27 days (days in vitro, DIV27) and found similar reductions in the levels of NRF-1, ATP5A, ATPB, and VDAC1 from cells of *Fmr1 KO* mice (Figure 6B), which was consistent with the reduced levels observed in the mitochondrial fractions. These results indicate that mitochondrial dysfunction could be monitored by measuring the depletion of mitochondrial components in EVs.

## 3. Discussion

The alterations of mitochondrial proteins and impaired energy homeostasis in the brains of *Fmr1 KO* mice suggest that mitochondrial dysfunction contributes to the pathogenesis of FXS [24,25,26]. In our study, levels of ATP synthase and mitochondrial membrane proteins were decreased in mitochondria from *Fmr1 KO* mouse brains, resulting in reduced MMPs in astrocytes. ATP synthase produces the energy necessary to maintain mitochondrial function, and other mitochondrial proteins are essential for regulating MMP generation and execution in mitochondria [27], thereby maintaining mitochondrial homeostasis. The permeability of the mitochondrial membrane protein VDAC is regulated by interactions with VIF to maintain the MMP [28]. The interaction of mitochondria with VIF stabilizes their intracellular location and activity [29]. We found that VDAC1 expression was decreased in the mitochondrial fraction from *Fmr1 KO* mouse astrocytes and that vimentin was more dispersed, observed at the endfeet of *Fmr1 KO* astrocytes, which may have contributed to the observed decrease in MMP.

Recent studies have highlighted that one of the most effective forms of communication between astrocytes and neurons occurs through EVs [30]. EVs secreted by astrocytes under normal conditions are well known to have neurotrophic and neuroprotective properties. Astrocytes-derived EVs under ischemic, oxidative stress, nutrient-deprived, or thermal stress conditions have been reported to carry various factors involved in increasing neuronal survival, guarding neurons against neurotransmitter toxicity, and promoting neurite outgrowth [31]. By contrast, under neurodegenerative disease conditions such as Alzheimer’s disease (AD), Parkinson’s disease (PD), and Amyotrophic Lateral Sclerosis (ALS), astrocyte-derived EVs have been suggested to contribute to the spread of neuropathology and the exacerbation of the extent of neurodegeneration [32,33]. Astrocytes are more resistant than neurons to acute external stresses, such as ischemia and hypoxia resulting from brain injury, and protect neuronal cell metabolism, ion balance, and signal transmission and even help in neuronal recovery [34,35,36]. Furthermore, mitochondria-derived from astrocytes rescue neurons whose mitochondria are damaged by stroke [10]. One mechanism by which they do this may be by secreting factors important for neurogenesis and neural function [37]. The treatment of culture medium of astrocytes from *Fmr1 KO* mice increases the percentage of large-size neurospheres but not the number of neurospheres, indicating that proteins secreted by astrocytes affect neural proliferation and differentiation [38,39,40]. However, chronic stress reduces the proliferation of astrocytes in the amygdala and not in the hippocampus [41]. Reduced proliferation is associated with a decrease in MMP and mitochondrial mass caused by loss of PTEN-induced kinase 1 (PINK1) [42]. This mitochondrial serine/threonine-protein kinase, encoded by the *PINK1* gene, protects cells from stress-induced mitochondrial dysfunction by binding with parkin and attaching to the depolarized mitochondria, causing autophagy. A previous study reported increased activation of astrocytes in *Fmr1 KO* mice, observed as increases in GFAP expression and the number of astrocytes in the corpus callosum and reactive astrogliosis in cultured primary astrocytes [43]. Similarly, astrocyte activation was also observed in the cerebella of *Fmr1 KO* mice [44]. In the present study, astrocytes from *Fmr1 KO* mice had reduced MMPs, which may be related to astrocytic activation and proliferation, but differs in vitro and in vivo. Further studies are needed to address the relationship between astrocyte activation and MMP in *Fmr1 KO* mice. 

Cortical neurons with reduced intracellular ATP and viability are rescued by EVs containing functional mitochondria secreted from primary astrocytes [45]. The reduced expression of mitochondrial components in mitochondrial fractions observed in the present study was accompanied by a reduction of these proteins in EVs that were secreted into the extracellular milieu. Liquid chromatography-mass spectrometry (LC-MS) analysis confirmed the decrease in mitochondrial proteins in EVs derived from *Fmr1 KO* mice (data not shown). These data support an EV-mediated transport of mitochondrial components. Other mechanisms have been proposed to underlie the transfer of mitochondria between cells, including membrane evulsions (during transcellular mitophagy) and tunneling nanotubes [46]. Before being exported to the extracellular environment, toxic, obsolete, or damaged mitochondrial material is loaded into endolysosomes for degradation of extracellular export via vesicles such as EVs. The presence of mitochondrial molecules in EVs is indirect evidence of the cross-talk between mitochondria and the endolysosomal system [47]. Indeed, protein levels in EVs are dynamically regulated under different conditions [48]. For example, target proteins are decreased in EVs when the target protein is fairly translated into protein resulting small amount of total protein in the cytosol, or the target proteins are attenuated in cytosol not to be released [49]. 

In the present study, astrocytes from *Fmr1 KO* mice had reduced amounts of mitochondrial proteins in mitochondrial fractions and EVs, including those important for mitochondrial biogenesis and transcriptional activities. However, these proteins were still detectable in moderate amounts in mitochondrial fractions compared with that in the EVs secreted into the culture medium. Thus, the mitochondrial proteins were possibly depleted from astrocytic EVs before their secretion with different mechanisms in neurons and astrocytes. Further studies are needed to determine the precise mechanism(s) for the observed decrease in mitochondrial proteins, which could reflect a deficit in the intracellular trafficking from mitochondria to EVs or disruption of EVs formation. Moreover, to understand better the complex physiological functions of astrocytes-derived EVs, it will be crucial to clarify which EV components influence the progression of FXS pathology and its regulatory mechanism.

In conclusion, we found that mitochondrial components are delivered through EVs but are diminished in EVs derived from cerebral cortices and those secreted from astrocytes of *Fmr1 KO* mice. The depletion of mitochondrial proteins accompanies a decrease in the MMP, thereby contributing to mitochondrial dysfunction in astrocytes. These results suggest that interplay between mitochondria and EVs reflects mitochondrial dysfunction of astrocytes associated with diseases. 

## 4. Materials and Methods 

### 4.1. Animal Maintenance 

All animal experiments were approved by the Institutional Animal Care & Use Committee of the Korea Brain Research Institute (21 February 2019) and was registered (IACUC-19-00012, 28 February 2019). C57BL/6J WT and B6.129P2-*Fmr1*^tm1Cgr^/J FXS model mice were purchased from the Jackson Laboratory. Mice were maintained on a C57Bl6/J background. Hemizygous male *Fmr1 KO* (*Fmr1*^−/y^) and wild-type (WT, *Fmr1*^+/y^) mice were used at postnatal day two (P2) and at ten weeks. Mice were housed in a specific-pathogen-free facility with 12 h of light and 12 h of dark per day at an ambient temperature of 22 °C and relative humidity of 40% ± 5%. Food and water were provided *ad libitum*.

### 4.2. Primary Astrocyte Cultures 

Cortical astrocytes were isolated from cerebral cortices dissected from mice at P2. Briefly, the cortices were isolated from P2 mice brain and dissociated for 20 min at 37 °C in Dulbecco’s modified Eagle medium (Thermo Fisher, Waltham, MA, USA), 0.1% trypsin-EDTA (GIBCO, MA, USA), 10% fetal bovine serum (Thermo Fisher), and 100 U/mL penicillin-streptomycin (GIBCO). After centrifugation at 2000 rpm for 1 min, cells were gently dissociated by pipetting and then filtered through a cell strainer. Cortical astrocytes were cultured in T75 flasks coated with poly-l-lysine for seven days. To eliminate microglia, neurons, and oligodendrocytes, astrocytes at confluency (at DIV7) were placed horizontally on a shaker platform with a medium covering the cells and shaken for 2 h at 350 rpm and then another 6 h after changing the medium. The supernatants were then removed, and astrocytes were transferred to a new T75 flask. Astrocytes were maintained for three weeks in culture before experimental use. 

### 4.3. qRT-PCR Analysis 

Total RNA was isolated from cortical samples using the RNeasy Mini kit according to the manufacturer’s instructions (Qiagen, Hilden, Germany), and 1 μg was used to synthesize the first strand of cDNA using the Superscript first-strand synthesis system for qRT-PCR (Invitrogen). qRT-PCRs were performed in triplicates using a FastStart SYBR green master mix in an ABI Prism 7300 sequence detection system (Applied Biosystems, Cummings Center, Beverly, MA, USA). The expression levels of *Nfe2l1*, *Nfe2l2*, and *Tfam* relative to that of the endogenous reference gene actin were calculated using the delta cycle threshold (ΔΔ*C_T_*) method. To assess mitochondrial biogenesis, as described in detail previously [50], the ratios of mitochondrial ribosomal RNA (16S) and *mt-Co1* to nuclear gene ribosomal protein large p0 (RPLP0) were quantified by qRT-PCR, assuming that RPLP0 levels remain constant. DNA was extracted from the cortex using a genomic DNA purification kit (Promega, WI, USA), according to the manufacturer’s instructions. SYBR green was used to measure expression levels, and data were normalized against the expression of RPLP0 (ΔΔ*C_T_* analysis). All primers are listed in Appendix A.

### 4.4. Mitochondria Fractionation 

Cortices (P2 mice) and astrocytes (DIV27) were resuspended in extraction buffer from the Mitochondria isolation kit (Thermo Fisher) according to the manufacturer’s instructions. After they were homogenized with a tissue grinder on ice, homogenates were centrifuged at 700× *g* for 10 min at 4 °C, and the supernatant was collected and centrifuged 12,000× *g* again for 15 min at 4 °C. The resulting supernatant and pellet were collected and considered cytosolic and mitochondrial fractions, respectively. Both fractions were analyzed with the BCA assay method (Pierce, MA, USA) and subjected to SDS-PAGE for western blot analysis.

### 4.5. EVs Isolation and Purification 

To isolate cortical EVs, sliced cortices of 10-week-old mice were predigested with 0.1% collagenase I (Worthington, NJ, USA) buffer including 0.001% DNase I (Sigma, St. Louis, MO, USA) and a protease inhibitor cocktail (Thermo Fisher) for 30 min at 37 °C in a hybridizer incubator. 

The digested cortical tissue was precipitated with 50% (*w/v*) polyethylene glycol 4000 (Merck, MO, USA) overnight at 4 °C and centrifuged at 12,000× *g* for 20 min at 4 °C. After removing the supernatant, the pellet was resuspended in HEPES-buffered saline (1 M HEPES, 5 M NaCl) and centrifuged again (12,000× *g*, 20 min). The pellet was once again resuspended in HEPES-buffered saline and then mixed with 50% (*w/v*) iodixanol (OptiPrep^TM^, Axis-Shield). Samples were purified by buoyant density gradient ultracentrifugation (200,000× *g*, 2 h, 4 °C.) through layers of 5%, 20%, and 30% iodixanol. The EV fraction was collected between the top and middle layers.

To isolate astrocyte EVs, the medium collected from astrocyte cultures was centrifuged at 400× *g* and 2000× *g* for 10 min, sequentially, and then filtered through a 0.25-μm syringe filter. Filtered samples were concentrated with a tangential flow filtration system with a 100 kDa molecular cutoff membrane (Pall Corporation, NY, USA). Concentrated samples were precipitated with 50% (*w/v*) polyethylene glycol 4000 (Merck) overnight at 4 °C and centrifuged at 12,000× *g* for 20 min at 4 °C, after which all procedures were the same as cortical EV isolation.

### 4.6. Nanoparticle Tracking Analysis (NTA)

The EVs samples were diluted 1:100 with HEPES-buffered saline, and 500 μL was loaded into the chamber of a NanoSight LM10 (Malvern Panalytical, Malvern, UK). The number of particles per milliliter was calculated by NanoSight NTA 3.2 software. After NTA, EVs were identified by transmission electron microscopy and western blot analysis.

### 4.7. Transmission Electron Microscopy (TEM)

For transmission electron microscopy, 400 mesh copper grids with carbon-coated formvar film (Electron Microscopy Sciences, EMS, PA, USA) were used. Six microliters of cortical EV samples (1/3 diluted with distilled water) were placed on a grid and incubated for 5 to 10 min. After soaking, grids were serially washed, stained with 1% uranyl acetate solution (EMS), and air-dried. All used solutions were filtered through a 0.22 μm filter (Merck). Images were acquired by a high-speed transmission electron microscope (Tecnai G2, FEI) at Brain Research Core Facilities in Korea Brain Research Institute.

### 4.8. Western Blot Analysis 

Isolated cortices and cultured cortical astrocytes were lysed using protein lysis buffer (25 mM Tris-HCl (pH 7.6), 150 mM NaCl, 1% NP-40, 1% sodium deoxycholate, and 0.1% SDS) with a protease and phosphatase inhibitor cocktail (Thermo Fisher). Protein concentrations were determined by the BCA assay method (Pierce). Protein samples were loaded on 8%–12% SDS-PAGE gels and then transferred to polyvinylidene fluoride membranes (Millipore, MO, USA). The membranes were blocked with 5% skim milk for 1 h and then analyzed by western blotting using the antibodies listed in Appendix A. The western blot images were acquired using a LAS 4000 imaging system (Fujifilm, Tokyo, Japan). For quantification, the images were scanned, and the intensities of protein bands were measured using NIH Image J software.

### 4.9. Measurement of MMP

The MMPs of astrocytes (DIV27) were determined by a MITO-ID^®^ membrane potential detection kit (Enzo, NY, USA) according to the manufacturer’s protocol. Briefly, astrocytes were stained with MITO-ID^®^ MP detection reagent solution for 30 min at 37 °C. Astrocytes incubated with potent mitochondrial oxidative phosphorylation uncoupler (40 μM 2-(2-[3-chlorophenyl]hydrazinylyidene) propanedinitrile) for 30 min were used as a negative control. After adding buffer B, astrocytes were imaged using an A1RSi confocal microscope system (Nikon) with a 40× lens objective. The MMP was assessed by quantifying the ratio of the intensity of orange fluorescence (emission wavelength 570 nm) to green fluorescence (emission wavelength 530 nm).

### 4.10. Immunocytochemistry 

Cultured primary cortical astrocytes (DIV27) were briefly washed with phosphate-buffered saline (PBS) and then fixed with 4% paraformaldehyde in PBS. The cells were permeabilized with 0.25% Triton X-100 in PBS and incubated with PBS containing 10% donkey serum for 1 h. The cells were then incubated with primary antibodies against vimentin (1:500) and GFAP (1:300) for 1 h at room temperature, followed by washing and incubation with Alexa Fluor 488- or Alexa Fluor 594- conjugated anti-IgG (Invitrogen, 1:500) for 1 h. After three washes (5 min each) using PBS, cells were mounted in Vectashield mounting medium with DAPI (Vector laboratories, CA, USA) for fluorescence analysis. The images were acquired using an A1RSi confocal microscope system (Nikon, Tokyo, Japan) equipped with a 40× lens objective.

### 4.11. Statistical Analysis

Statistical significance was evaluated by an unpaired two-tailed *t*-test. All values are expressed as the mean ± SD. 

## Figures and Tables

**Figure 1 ijms-22-00410-f001:**
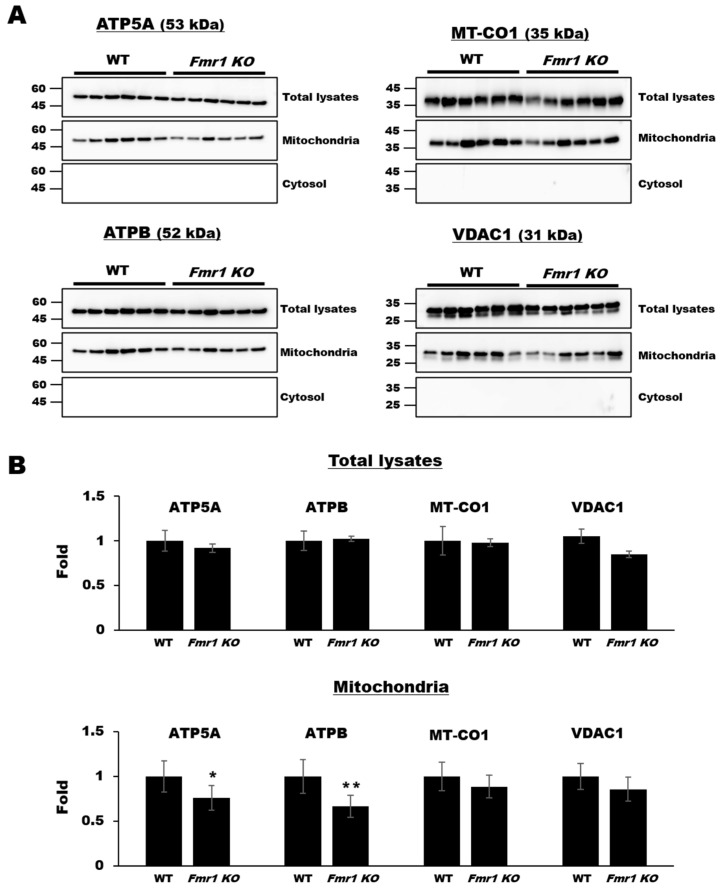
Decreased expression of mitochondrial functional components in mitochondrial fractions but not cortical lysates from *Fmr1 knockout (KO)* mice. (**A**) Western blot analysis of ATP5A, ATPB, MT-CO1, and VDAC1 in cortical lysates, mitochondrial fraction, and cytosol fraction of cortices in WT or *Fmr1 KO* mice. (**B**) Expression levels were determined by measuring band densities with NIH ImageJ software. Each value represents the mean ± SD (*n* = 6 per group). * *p* < 0.05; ** *p* < 0.01 vs. WT mice.

**Figure 2 ijms-22-00410-f002:**
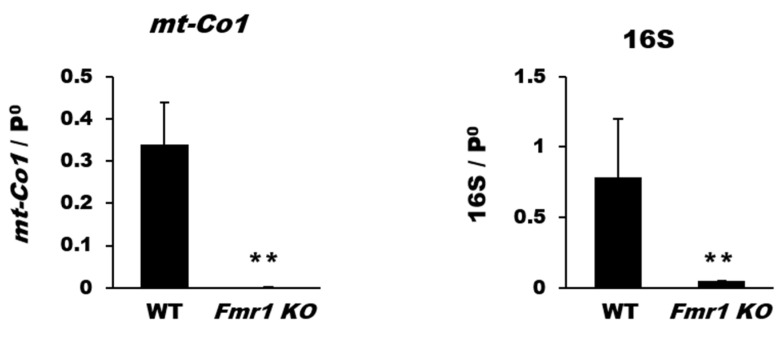
Loss of genes coding for mitochondrial biogenesis in *Fmr1 KO* cortex. Gene expression of markers of mitochondrial biogenesis was determined by qRT-PCR. The relative amounts of mtDNA (*mt-Co1* and 16S) and nuclear DNA (p0) were compared. Each value represents the mean ± SD of three mice. ** *p* < 0.01 vs. WT mice.

**Figure 3 ijms-22-00410-f003:**
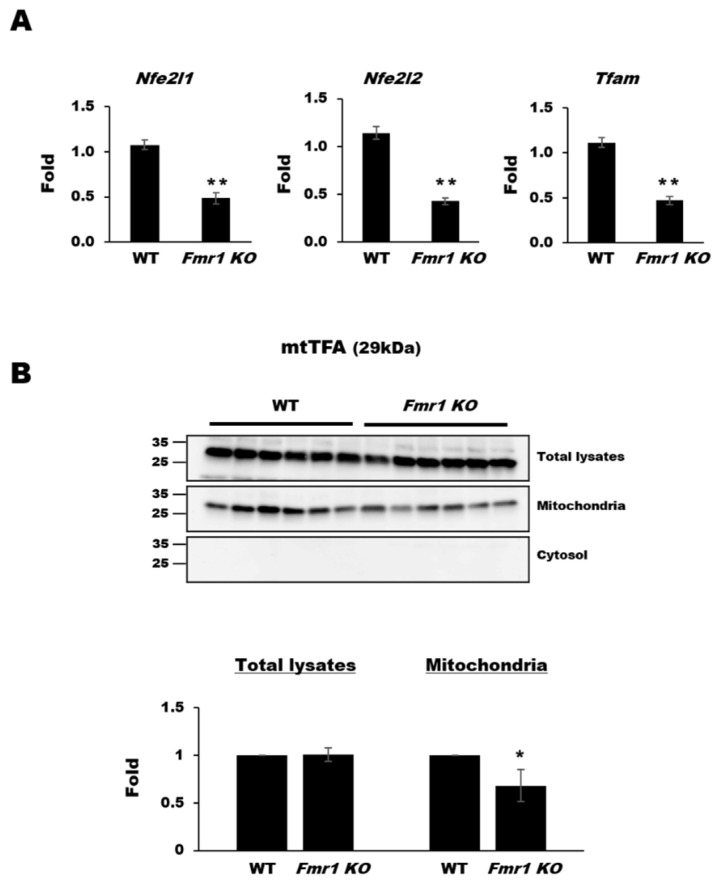
Reduction in mitochondrial transcriptional components in the *Fmr1 KO* cortex. (**A**) The relative expression levels of *Nfe2l1*, *Nfe2l2*, and *Tfam* are expressed as the mean ± SD of three independent experiments. ** *p* < 0.01 vs. *Fmr1 KO* mice. (**B**) Representative western blots and quantitative analysis of mitochondrial transcription factor A (mtTFA). Each value represents the mean ± SD (*n* = 6 per group). * *p* < 0.05 vs. WT mice.

**Figure 4 ijms-22-00410-f004:**
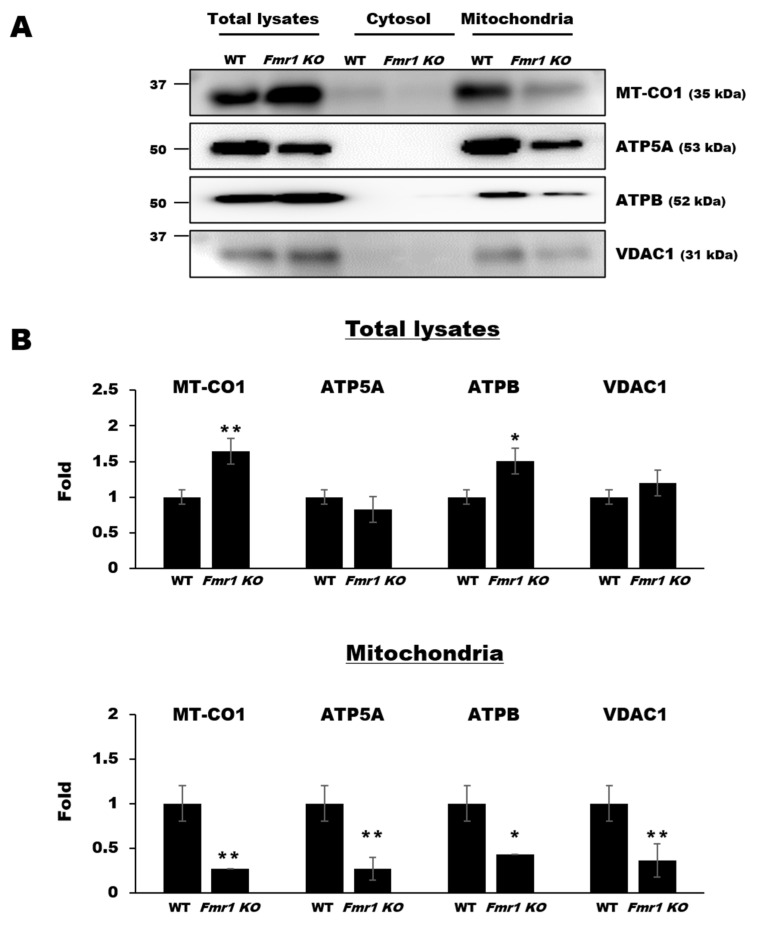
Decreased protein expression in the astrocyte mitochondrial fractions of *Fmr1 KO* mice. Representative western blots (**A**) and relative quantification (**B**) of the expression of MT-CO1, ATP5A, ATPB, and VDAC1 in total lysates, cytosol, and mitochondrial fractions from astrocytes of WT and *Fmr1 KO* mice. Expression levels were determined by measuring band densities with the National Institute of Health (NIH) ImageJ software. Each value represents the mean ± SD of three independent experiments. * *p* < 0.05; ** *p* < 0.01 vs. WT astrocytes.

**Figure 5 ijms-22-00410-f005:**
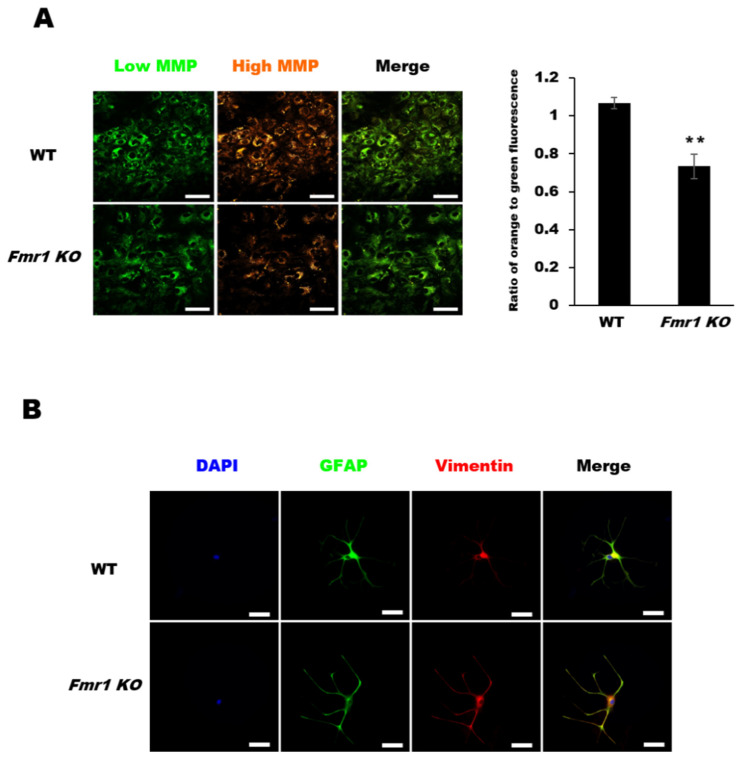
Mitochondrial membrane potentials (MMPs) and vimentin distribution in astrocytes of *Fmr1 KO*. (**A**) The mitochondrial membrane potential of astrocytes was determined using MITO-ID^®^, which contains a cationic dye that fluoresces green at low mitochondrial membrane potential (MMP) and orange at high MMP. Astrocytes (DIV27) were incubated with MITO-ID^®^ for 30 min and then visualized by confocal microscopy (left). Astrocytes from *Fmr1 KO* mice showed lower MMP than astrocytes from WT mice (right). Representative images are shown at 400× magnification. Each value represents the mean ± SD (*n* = 8 per group). ** *p* < 0.01 vs. WT astrocytes. Scale bars, 50 μm. (**B**) Double immunofluorescence staining with anti-GFAP and anti-vimentin in astrocytes from WT and *Fmr1 KO* mice. Astrocytes of *Fmr1 KO* showed abnormal localization of the vimentin intermediate filament (VIF). Scale bars, 50 μm.

**Figure 6 ijms-22-00410-f006:**
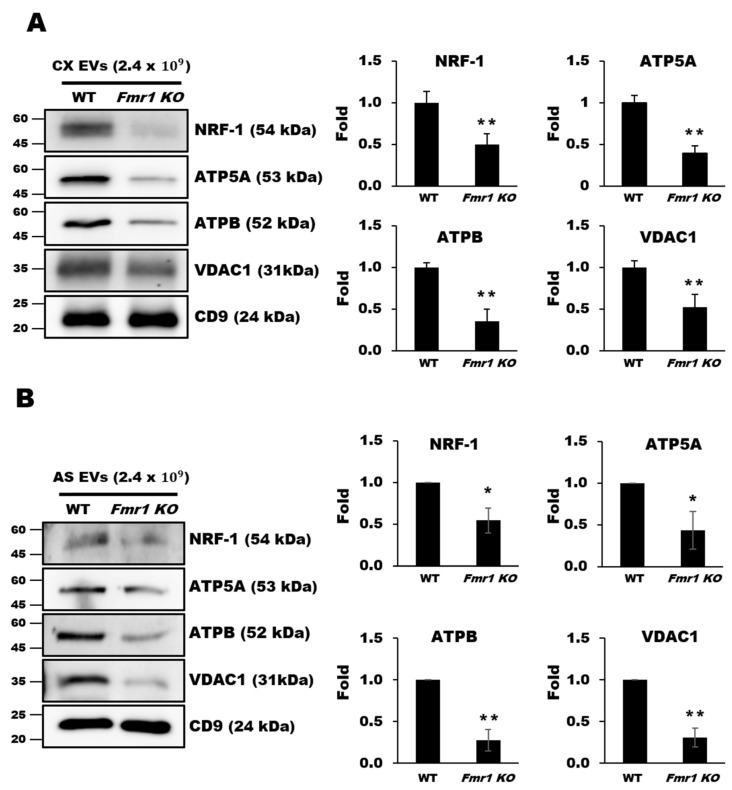
Depletion of mitochondrial components in extracellular vesicles (EVs) from cerebral cortices (**A**) and astrocytes (**B**) of *Fmr1 KO* mice. The expression of mitochondrial proteins including NRF1, ATP5A, ATPB, and VDAC1 was analyzed by western blot analysis of EVs isolated from the cortices (CX) (**A**) and astrocytes (AS) (**B**) of WT and *Fmr1 KO* mice. Fold differences in expression were determined by measuring band densities with NIH ImageJ software. CD9 was used as an EVs marker and loading control. Each value represents the mean ± SD of three independent experiments. * *p* < 0.05; ** *p* < 0.01 vs. WT mice.

## Data Availability

Data available in a publicly accessible repository.

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
