# Peer review of "Depletion of Mitochondrial Components from Extracellular Vesicles Secreted from Astrocytes in a Mouse Model of Fragile X Syndrome"

_ijms, 2021, doi:10.3390/ijms22010410_

Round 1

Reviewer 1 Report

The article "Depletion of mitochondrial components from extracellular vesicles secreted from astrocytes in a mouse model of Fragile X syndrome" is an interesting article, which suggest that mitochondrial dysfunction in astrocytes is associated with the pathogenesis of Fragile X syndrome and can be monitored by a depletion of components in EVs.

Major points

Figure 1. Decreased expression of mitochondrial functional validated by western blot analysis and band densities quantification. Would be great if the author do some real time PCR or other assays to prove the concept. Only one gene and 16s is done in figure 2.

Similarly in figure 3. reduction in mitochondrial transcriptional components in Fmr1 KO cortex is done by relative expression levels of Nfe2l1, Nfe2l2, and Tfam. If possible the author is encouraged to perform flow cyotometry for these proteins.

If possible the author is encouraged for more functional assay (mitochondrial influx, etc assays) of mitochondrial functional analysis.

Author Response

The article "Depletion of mitochondrial components from extracellular vesicles secreted from astrocytes in a mouse model of Fragile X syndrome" is an interesting article, which suggest that mitochondrial dysfunction in astrocytes is associated with the pathogenesis of Fragile X syndrome and can be monitored by a depletion of components in EVs.

Major points

Figure 1. Decreased expression of mitochondrial functional validated by western blot analysis and band densities quantification. Would be great if the author do some real time PCR or other assays to prove the concept. Only one gene and 16s is done in figure 2.

We also agree with the reviewer’s suggestion and apologize for not presenting many results in this manuscript. We would like to briefly introduce our unpublished data instead of conducting new experiments in a limited time. qRT-PCR results showed that the mRNA expression patterns of mitochondrial components such as ATP5A and ATPB are decreased but not significant. VDAC1 is not changed, which is similar to the protein expression pattern in the cortex (Figure 1). It would be good to include these results in the current manuscript. However, we still need to dissect the mechanisms of the translation and trafficking of mitochondrial components in further studies. We preliminarily found that Fmr1 KO mice show the different expression patterns of mitochondrial components in astrocytes vs. neuronal cells (data not shown). We are currently investigating which mechanisms induce this phenomenon through various experiments such as proteome analysis, immunocytochemistry, functional analysis, etc.

Figure.1 Relative mRNA expressions of mitochondrial components in the cortex of Fmr1 KO mice (please see attached file)

In the next research project, we hope that it can be present the conclusion through various analysis methods as the reviewer’s suggest.

Similarly in figure 3. reduction in mitochondrial transcriptional components in Fmr1 KO cortex is done by relative expression levels of Nfe2l1, Nfe2l2, and Tfam. If possible the author is encouraged to perform flow cyotometry for these proteins.

We appreciate the valuable point.

We attempted to do flow cytometry to see the transcription factors, but it is hard to detect nuclear proteins such as NRF-1 and NRF-2 (encoded by Nfe2l1 and Nfe2l2, respectively) using flow cytometry. Besides, because of the limited time to obtain the brain samples, P2 brain of Fmr1 KO mice, we tried to do the western blot in adult brains showing that NRF-1 protein was reduced in Fmr1 KO cortex as shown in Figure 2. We were afraid to include this figure in the main figures because of age mismatch so that we may include this figure in supplement with the agreement of reviewer 1.

Figure 2. Expression of NRF-1 in the cortex of 10 weeks old Fmr1 KO mice.

Regarding the function of transcription factors such as NRF-1 and 2 related with mitochondrial function, it has been reported that NRF-1 and NRF-2 mutant mice result in late gestational embryonic lethality and impaired oxidative stress defense, indicating Nrf1 and 2 deficiency is related to oxidative stress causing the impaired mitochondrial transcriptional system and mitochondrial dysfunction [1, 2].

Reference

  1. Leung L, Kwong M, Hou S, Lee C, Chan JY. Deficiency of the Nrf1 and Nrf2 transcription factors results in early embryonic lethality and severe oxidative stress. J Biol Chem. 2003 Nov 28;278(48):48021-9. doi: 10.1074/jbc.M308439200.
  2. Fmrl Knockout Mice: A Model to Study Fragile X Mental Retardation Cell, Vol. 78, 23-33, July 15, 1994

If possible the author is encouraged for more functional assay (mitochondrial influx, etc assays) of mitochondrial functional analysis.

We have attempted to do mitochondrial functional analysis, but it takes time to obtain the kit because of pandemic circumstances. Instead, we tried to see the localization and expression patterns of mitochondrial membrane protein, VDAC1, and ATP synthase, ATP5A using immunostaining. They are reduced in overall astrocytes and mostly localized near the nucleus area of astrocytes derived from Fmr1 KO mice. It is expected that abnormal distribution and diminished expression of mitochondrial proteins leads the mitochondrial dysfunction in Fmr1 KO mice. We also agree that our results are insufficient to conclude the protein deficiency is correlated with mitochondrial dysfunction. We will furtherly study on the differential mechanisms of mitochondrial dysfunction in neurons vs.. astrocyte due to their discrepancy expression profiles of mitochondrial proteins as we mentioned in response on first comment of reviewer 1. It will be included like mitochondrial enzyme activity analysis such as OXPHOS activity (including complex I, II, III, IV, and V and citrate synthase activity.

Reviewer 2 Report

The paper presented ihere s original in many respects. It is focused on mitochondria, structures of great importance that are only partially known in exchange cell biology; and in alterations of the EV functins appearing in FXS, a transcriptional defect genetic disease, appeared in astrocytes, important glia cells of the brain, unknown in the disease. The paper is widely investigated. Therefore various results obtained are interesting to read. However, their scientific soundness is to be controlled. Many data, received as hypotheses, are presented as critical data; others, such as the definition of the EV cargo generation, are given as solid while their occurrence it is variable depending on the time of cell function. In other words the paper needs to be re-considered critically, considering also recent developments on astrocytes influencing  the field illustrated.

Author Response

Reviewer 2

Comments and Suggestions for Authors

The paper presented ihere s original in many respects. It is focused on mitochondria, structures of great importance that are only partially known in exchange cell biology; and in alterations of the EV functins appearing in FXS, a transcriptional defect genetic disease, appeared in astrocytes, important glia cells of the brain, unknown in the disease. The paper is widely investigated. Therefore various results obtained are interesting to read. However, their scientific soundness is to be controlled. Many data, received as hypotheses, are presented as critical data; others, such as the definition of the EV cargo generation, are given as solid while their occurrence it is variable depending on the time of cell function. In other words the paper needs to be re-considered critically, considering also recent developments on astrocytes influencing the field illustrated.

According to the reviewer’s suggestion, we have now added more information in Introduction (line 65-67, page 2) and Discussion (line 195-203, page 16; line 246-248, page 17).

Recent studies have highlighted that one of the significant communication forms between astrocytes and neurons occurs through extracellular vesicles (EVs) [1–5].

The EVs secreted by astrocytes in normal conditions are believed to have neurotrophic and neuroprotective properties, whereas EVs released from astrocytes in ischemic, oxidative stress, nutrient-deprived, or thermal stress conditions have been shown to carry various factors that are involved in neuronal survival, guarding neurons against neurotransmitter toxicity, and promoting neurite outgrowth [6–10]. However, in neurodegenerative disease conditions such as Alzheimer's disease (AD), Parkinson's disease (PD), and Amyotrophic Lateral Sclerosis (ALS), astrocyte-derived EVs (ADEVs) have been suggested to have a role in spreading neuropathology or exacerbating the extent of neurodegeneration [11, 12].

Based on studies until now, although the physiological role and regulatory mechanism of EVs in FXS are not yet unknown, at least it is suggested that it can be applied to diagnosis through EVs.

References

[1] F. Bianco, C. Perrotta, L. Novellino, M. Francolini, L. Riganti, E. Menna, L. Saglietti, E.H. Schuchman, R. Furlan, E. Clementi, M. Matteoli, C. Verderio, Acid sphingomyelinase activity triggers microparticle release from glial cells, EMBO J. 28 (2009) 1043–1054.

[2] A. Datta Chaudhuri, R.M. Dasgheyb, L.R. DeVine, H. Bi, R.N. Cole, N.J. Haughey, Stimulus-dependent modifications in astrocyte-derived extracellular vesicle cargo regulate neuronal excitability, Glia 68 (2020) 128–144.

[3] C. Fruhbeis, D. Frohlich, W.P. Kuo, J. Amphornrat, S. Thilemann, A.S. Saab, F. Kirchhoff, W. Mobius, S. Goebbels, K.A. Nave, A. Schneider, M. Simons, M. Klugmann, J. Trotter, E.M. Kramer-Albers, Neurotransmitter-triggered transfer of exosomes mediates oligodendrocyte-neuron communication, PLoS Biol. 11 (2013) e1001604.

[4] M. Guescini, S. Genedani, V. Stocchi, L.F. Agnati, Astrocytes and glioblastoma cells release exosomes carrying mtDNA, J. Neural Transm. (Vienna) 117 (2010) 1–4.

[5] A.R. Taylor, M.B. Robinson, D.J. Gifondorwa, M. Tytell, C.E. Milligan, Regulation of heat shock protein 70 release in astrocytes: role of signaling kinases, Dev. Neurobiol. 67 (2007) 1815–1829.

[6] A.R. Taylor, M.B. Robinson, D.J. Gifondorwa, M. Tytell, C.E. Milligan, Regulation

of heat shock protein 70 release in astrocytes: role of signaling kinases, Dev. Neurobiol. 67 (2007) 1815–1829.

[7] R.D. Gosselin, P. Meylan, I. Decosterd, Extracellular microvesicles from astrocytes contain functional glutamate transporters: regulation by protein kinase C and cell activation, Front. Cell. Neurosci. 7 (2013) 251.

[8] S. Moidunny, J. Vinet, E. Wesseling, J. Bijzet, C.H. Shieh, S.C. van Ijzendoorn, P. Bezzi, H.W. Boddeke, K. Biber, Adenosine A2B receptor-mediated leukemia inhibitory factor release from astrocytes protects cortical neurons against excitotoxicity, J. Neuroinflammation 9 (2012) 198.

[9] P. Proia, G. Schiera, M. Mineo, A.M. Ingrassia, G. Santoro, G. Savettieri, I. Di Liegro, Astrocytes shed extracellular vesicles that contain fibroblast growth factor- 2 and vascular endothelial growth factor, Int. J. Mol. Med. 21 (2008) 63–67.

[10] S. Wang, F. Cesca, G. Loers, M. Schweizer, F. Buck, F. Benfenati, M. Schachner, R. Kleene, Synapsin I is an oligomannose-carrying glycoprotein, acts as an oligomannose-binding lectin, and promotes neurite outgrowth and neuronal survival when released via glia-derived exosomes, J. Neurosci. 31 (2011) 7275–7290.

[11] G. Schiera, C.M. Di Liegro, I. Di Liegro, Extracellular membrane vesicles as vehicles for brain cell-to-cell interactions in physiological as well as pathological conditions, Biomed. Res. Int. 2015 (2015) 152926.

[12] J.M. Silverman, D. Christy, C.C. Shyu, K.M. Moon, S. Fernando, Z. Gidden, C.M. Cowan, Y. Ban, R.G. Stacey, L.I. Grad, L. McAlary, I.R. Mackenzie, L.J. Foster, N.R. Cashman, CNS-derived extracellular vesicles from superoxide dismutase 1 (SOD1)(G93A) ALS mice originate from astrocytes and neurons and carry misfolded SOD1, J. Biol. Chem. 294 (2019) 3744–3759.

Round 2

Reviewer 1 Report

No comments. I think the comments in my previous draft can improve the manuscript quality.

Author Response

We first apologize for not being able to respond sufficiently to reviewers.

It was complicated to add the result of functional analysis in a limited time (1st revision 10 days and 2nd revision 7 days) because, as indicated in the experimental method, at least 27 days of cell culture time were required to obtain astrocytes for mitochondrial function study. Therefore, we conducted experiments using the limited materials available and provided it as supplemental data as the reviewers requested (Figure S2 and S3). The results are described on page 2, line 83-84; page 4, line 109.

In our current study, we suggested that EVs carry the mitochondrial components to be secreted into the extracellular environment, which enables us to monitor the diseases associated with the mitochondrial dysfunction showing the abnormal expression of mitochondria protein. We completely agree to define the mitochondria dysfunction using the functional assay. In a viewpoint of mitochondrial function in FXS, several studies showed that aberrant cellular metabolism by dysfunctional mitochondrial proteins such as ATP synthase c-subunit leak [1] and coenzyme Q (CoQ) deficiency [2] was contributed to FXS phenotype. Moreover, a study about mitochondrial bioenergetics also suggested that altered mitochondrial energy metabolism, including the activity of key glycolytic enzymes, glycerol-3-phosphate shuttle, and mitochondrial respiratory chain (MRC) complexes may contribute to neurological impairment in FXS [3]. However, because it shows different results depending on the cell types and the age group [4, 5], the study on the mitochondria function must be conducted carefully, as the reviewers commented. For now, we are planning to extend the functional studies, including OXPHOS related enzyme activity, glycolytic enzyme activity, etc., in astrocytes and neurons as well as in the brain of embryonic (E16.5), neonatal (P2-3), and adult stage (3month).

References.

  1. Licznerski P, Park HA, Rolyan H, Chen R, Mnatsakanyan N, Miranda P, Graham M, Wu J, Cruz-Reyes N, Mehta N, Sohail S, Salcedo J, Song E, Effman C, Effman S, Brandao L, Xu GN, Braker A, Gribkoff VK, Levy RJ, Jonas EA. ATP Synthase c-Subunit Leak Causes Aberrant Cellular Metabolism in Fragile X Syndrome. Cell. 2020 Sep 3;182(5):1170-1185.e9. doi: 10.1016/j.cell.2020.07.008. Epub 2020 Aug 13. PMID: 32795412; PMCID: PMC7484101.
  2. Griffiths KK, Wang A, Wang L, Tracey M, Kleiner G, Quinzii CM, Sun L, Yang G, Perez-Zoghbi JF, Licznerski P, Yang M, Jonas EA, Levy RJ. Inefficient thermogenic mitochondrial respiration due to futile proton leak in a mouse model of fragile X syndrome. FASEB J. 2020 Jun;34(6):7404-7426. doi: 10.1096/fj.202000283RR. Epub 2020 Apr 20. PMID: 32307754; PMCID: PMC7692004.
  3. D'Antoni S, de Bari L, Valenti D, Borro M, Bonaccorso CM, Simmaco M, Vacca RA, Catania MV. Aberrant mitochondrial bioenergetics in the cerebral cortex of the Fmr1 knockout mouse model of fragile X syndrome. Biol Chem. 2020 Mar 26;401(4):497-503. doi: 10.1515/hsz-2019-0221. PMID: 31702995.
  4. Fecher C, Trovò L, Müller SA, Snaidero N, Wettmarshausen J, Heink S, Ortiz O, Wagner I, Kühn R, Hartmann J, Karl RM, Konnerth A, CKorn T, Wurst W, Merkler D, Lichtenthaler SF, Perocchi F, Misgeld T. Cell-type-specific profiling of brain mitochondria reveals functional and molecular diversity. Nat Neurosci. 2019 Oct;22(10):1731-1742. doi: 10.1038/s41593-019-0479-z. Epub 2019 Sep 9. PMID: 31501572.
  5. Elfawy HA, Das B. Crosstalk between mitochondrial dysfunction, oxidative stress, and age related neurodegenerative disease: Etiologies and therapeutic strategies. Life Sci. 2019 Feb 1;218:165-184. doi: 10.1016/j.lfs.2018.12.029. Epub 2018 Dec 20. PMID: 30578866.

Reviewer 2 Report

The changes introduced in te paper have improved the whole story, also because its is acknowledged that introduction of additional data, destined to be published elsewhere, would have been important also here. For this future publication I recommend a careful study of the papers published really recently, i.e. during the last 1-2 years, that often report significant changes of previous results.

Author Response

Thank you for pointing out several essential points that we overlooked in analyzing our current research results.

As a response to the comment of reviewer 1, we are planning to study on the mitochondrial function very carefully, which includes the functions studies such as OXPHOS related enzyme activity, glycolytic enzyme activity, etc., in astrocytes and neurons as well as in the brain of embryonic (E16.5), neonatal (P2-3), and adult stage (3month).